# Transition to adult care: Exploring factors associated with transition readiness among adolescents and young people in adolescent ART clinics in Uganda

**Scovia Nalugo Mbalinda**[1]*, **Sabrina Bakeera-Kitaka**[2], **Derrick Amooti Lusota**[1], **Philippa Musoke**[2], **Mathew Nyashanu**[1¤], **Dan Kabonge Kaye**[3]

1 Department of Nursing, College of Health Sciences, Makerere University, Kampala, Uganda,
2 Department of Peadiatrics, College of Health Science, Makerere University, Kampala, Uganda,
3 Department of Obstetrics and Gynaecology, College of Health Science, Makerere University, Kampala, Uganda

¤ Current address: Department of Health & Allied Professions, School of Social Sciences, Nottingham Trent University, Nottingham, United Kingdom
* snmbalinda@gmail.com

**Data Availability Statement:** All relevant data are within the manuscript and its Supporting Information files.

## Abstract

### Background

Transition readiness refers to a client who knows about his/her illness and oriented towards future goals and hopes, shows skills needed to negotiate healthcare, and can assume responsibility for his/ her treatment, and participate in decision-making that ensures uninter-rupted care during and after the care transition to adult HIV care. There is a paucity of research on effective transition strategies. This study explored factors associated with ado-lescent readiness for the transition into adult care in Uganda.

### Methods

A cross-sectional study was conducted among 786 adolescents, and young people living with HIV randomly selected from 9 antiretroviral therapy clinics, utilizing a structured ques-tionnaire. The readiness level was determined using a pre-existing scale from the Ministry of Health, and adolescents were categorized as ready or not ready for the transition. Bivariate and multivariate analyses were conducted.

### Results

A total of 786 adolescents were included in this study. The mean age of participants was 17.48 years (SD = 4). The majority of the participants, 484 (61.6%), were females. Most of the participants, 363 (46.2%), had no education. The majority of the participants, 549 (69.8%), were on first-line treatment. Multivariate logistic regression analysis found that readiness to transition into adult care remained significantly associated with having acquired a tertiary education (AOR 4.535, 95% CI 1.243–16.546, $P$ = 0.022), trusting peer educators for HIV treatment (AOR 16.222, 95% CI 1.835–143.412, P = 0.012), having received

**Funding:** Grant Number D43TW010132 supported by the Office of the Director, National Institutes of Health (OD), National Institute of Dental & Craniofacial Research (NIDCR), National Institute of Neurological Disorders and Stroke (NINDS), National Heart, Lung, And Blood Institute (NHLBI), Fogarty International Center (FIC), National Institute On Minority Health and Health Disparities (NIMHD). Received by Scovia Nalugo Mbalinda (SNM). Consortium for Advanced Research Training in Africa (CARTA). CARTA is jointly led by the African Population and Health Research Center and the University of the Witwatersrand and funded by the Carnegie Corporation of New York (Grant No–B 8606.R02), Sida (Grant No:54100029), the DELTAS Africa Initiative (Grant No: 107768/Z/15/Z). The DELTAS Africa Initiative is an independent funding scheme of the African Academy of Sciences (AAS) 's Alliance for Accelerating Excellence in Science in Africa (AESA) and supported by the New Partnership for Africa's Development Planning and Coordinating Agency (NEPAD Agency) with funding from the Wellcome Trust (UK) and the UK government. Received by Scovia Nalugo Mbalinda (SNM). The research reported in this publication was supported by the Fogarty International Center of the National Institutes of Health under Award Number 1R25TW011213. The content is solely the responsibility of the authors and does not necessarily represent the official views of the National Institutes of Health.

**Competing interests:** The authors have declared that no competing interests exist.

counselling on transition to adult services (AOR 2.349, 95% CI 1.004–5.495, $P$ = 0.049), having visited an adult clinic to prepare for transition (AOR 6.616, 95% CI 2.435–17.987, P = < 0.001) and being satisfied with the transition process in general (AOR 0.213, 95% CI 0.069–0.658, P = 0.007).

## Conclusion

The perceived readiness to transition care among young adults was low. A series of individual, social and health system and services factors may determine successful transition readiness among adolescents in Uganda. Transition readiness may be enhanced by strengthening the implementation of age-appropriate and individualized case management transition at all sites while creating supportive family, peer, and healthcare environments.

## Introduction

Perinatally or behaviorally, HIV-infected adolescents (10–19 years) and young adults (20–24 years) are an increasing proportion of the HIV-infected population in Uganda [1]. Globally, 1.8 million adolescents (10–19 years) live with HIV, and 85% of them live in sub-Saharan Africa. In Uganda [2]. In Uganda, 110,000 adolescents are living with HIV [3]. An increasing number of adolescents and young adults with HIV who require the transfer of care from pediatric to adult providers [4]; yet, globally, many young people experience barriers (e.g., infrastructure, staff training) that complicate this process. Transitioning from pediatric to adult HIV care has to progressively empower and increase the responsibility of HIV-infected adolescents and young people living with HIV (YPLHIV) to take charge of their health. A poor transition put the patients at risk of several challenges, including loss of continuity of care and stigma, both of which could lead to decreased adherence to care in general and ART in particular, with increased risk of HIV drug resistance and lowered immunity (and therefore morbidity from opportunistic infections) and risk of HIV transmission especially as the adolescents become sexually active. Such a transition nurtures adolescents' (and youth's) confidence, autonomy, and responsibility for their HIV care.

Transition readiness refers to the following client characteristics: 1) Client knows about his/her illness and oriented towards future goals and hopes, including long-term survival. 2) Client has the skills needed to negotiate appointments and multiple providers in an adult practice setting. 3) Client has personal and medical independence and can assume responsibility for his/her treatment and participate in decision-making. 4) Client is active and has been receiving uninterrupted care. 5) Client's basic and psychosocial needs, such as housing, employment, education, home-based services, or transportation, have also been addressed. 6) Client is familiar with the new providers and setting and has participated in discussions of the transition plan for themselves [14]. With more HIV-infected children and adolescents surviving into adulthood, these young patients' transition into adult care should be a top priority for health services [5]. For adolescents living with HIV, the transition should be a purposeful, planned movement from paediatric, child-centered, or specialized adolescent services to adult-oriented services. It is an important factor in the long-term health and well-being of an adolescent [5]. Recent advances in adolescent and young adult health care provide opportunities to optimize HIV prevention and care research outcomes [6, 7].

The Ministry of Health (Uganda) has recognized the importance of the HIV care transition. There is a standard protocol from the Ministry of Health to guide the transition; however, the

extent to which it is used is unknown, and there are only limited data on successful transition for Ugandan YPLHIV. A national survey of adolescent HIV services identified only one transition clinic at a National Referral Hospital and not readily accessible to 95% of the population living outside of Kampala [8–10]. Further, those at the highest risk for health complications and transmission of HIV to others (males, those living further away from health facilities) are among the least likely to have resources to support a successful transition [11–13].

The Ministry of Health has created an 8-page manual for adolescent HIV treatment and support for HIV providers to highlight the transition process's importance [14]. This document provides four key messages for transition: Develop a transition plan several years before the transition and update it at regular intervals, Ensure that HIV+ adolescents understand their chronic illness and its management; provide skills to negotiate adults clinic settings, Assess adolescents in an individualized manner for skills development and understanding of successful transition and Address barriers for each patient that may prevent skills acquisition (e.g., developmental delays, anxiety, PTSD) and advises providers to develop a transition plan and then implement that plan with no specific guidance on how to do so effectively. However, the extent of the implementation of these guidelines is not known. In this period, the provision of HIV care is a continuum, where some adolescents attend HIV care in adult clinics. At the same time, some young people still seek HIV care in adolescent clinics, potentially affecting the quality of care received. There need to assess HIV care transition readiness for the population of adolescents and young people. Given the low-resource context, it is important to assess the factors associated with readiness to transition to adult clinics. This information generated may inform the development of policies, programs, and practices to improve the transition process. This process should start early and is planned and managed carefully, with good communication between caregivers, health providers, and the adolescent, and involving the adolescent in all the decisions taken around their ongoing care.

## Methods

### Study design and setting

This was a cross-sectional study of PHIV adolescents and young people in antiretroviral therapy (ART) clinics in four Uganda regions (Eastern, Western, and Northern). Data were collected from August 1, 2019, to January 30, 2020.

### Study population

Adolescents and young people living with HIV were selected from adolescent ART clinics through a consecutive sampling procedure. At each site, a research assistant recruited all the available participants in the ART clinic and enrolled those who fulfilled the inclusion criteria (Not ill, above ten years), and gave informed consent/assent for participation until the sample size was obtained. Trained research assistants conducted interviews with the participants in the absence of their parents or guardians.

### Sample size determination

The sample size was powered to determine the readiness of transitioning. The sample size was calculated using a prevalence of 50% of readiness to transition, a 95% confidence interval, and an error margin of 5%. The prevalence of 50% was used since there was no previous study done in our context to determine the readiness to transition. A design effect of 2 was used to give us 784 adolescents.

## Variables and measurements

Data were collected through face-to-face interviewer-administered questionnaires in a private room. The data collected from participants included socio-demographic characteristics (age at last birthday, gender, living status, main caregiver, education level, religion, marital status, being orphaned, and occupation), clinical and immunological data (the type of ART site, type of current treatment CD4, and viral load), HIV knowledge and disclosure, Transition process (Knowledge of own health, Knowledge of responsible behavior, Knowledge of response of emergency care, Knowledge of how to manage health care needs, Demonstration of responsible sexual behavior, keeping track of health needs, support groups, and transition plan). Ministry of Health developed this tool, and it is used to transition these adolescents in the ART clinic [14]. The readiness to transitioning was measured using variables in the transitioning process instrument. The total score of the tool was 38. If the young person scored 30 and above, they were ready to be transitioned [14]. The primary outcome measured was ready to transition if they scored 30 and above and not ready to transition if they scored below 30.

## Data management and analysis

Participant's age was clustered into 10–14 years and 15–19 years, 20–24, and above 24 years. Education status was grouped into out of school and in school, and education level was grouped into three categories: secondary, primary, and no education, and The occupation was grouped into three categories: students, unemployed and employed.

The proportion of those ready to transition was computed. To assess factors associated with readiness to transition, Pearson's chi-square and Student t-test were used to measuring the association between these variables and for categorical and continuous explanatory variables, respectively.

Stepwise logistic regression models were built to identify independent predictors of transition readiness. During model development, the researcher considered all predictor variables with a p-value of ≤0.2 [24] at bivariate analysis were considered for inclusion in the multivariate logistic regression model. Collinearity was assessed using a correlation matrix and crosschecked by using the variance inflation factor, which was set at 10 [25]. In case two variables were associated ($P < 0.05$), the variable explaining the largest variability (smaller p-value at univariate analysis) was retained. Significance was set at 0.05, and all of the analyses were two-tailed. Analyses were done using STATA®16.

## Ethical review and approval

Ethical reviews and approval were obtained from the Research Ethics Committee of School of Health Sciences, College of Health Sciences at Makerere University #SHSREC REF NO: 2019–029 and Uganda National Council of Science Technology (SS5063). Administrative clearance and permissions were also obtained from the management of each of the health facilities. Written informed consent was obtained from young people above 18 years. For adolescents below 18 years, assent from the adolescents and consent from parents or guardians was obtained and assent from the adolescents. Participation was voluntary, and all the interviews were conducted in private settings to ensure the participant's confidentiality.

## Results

### Socio-demographic characteristics of the participants

A total of 786 adolescents were included in this study. The mean age of participants was 17.48 years (SD = 4). The majority of the participants, 484 (61.6%), were females, most of the

participants 363(46.2%) had no education; 310(39.4%), 183(23.3%) of the participants were employed, and 24 0(30.5%) came from families that received social support for their health. Only 362(46.1%) we're living with their parents, and the rest were living with relatives 115 (14.6%) and others with grandparents 93(11.3%). Main daily caregivers included parents 360 (45.8%), relatives 107 (13.6%) and grandparents 100(12.7%). **Table 1**.

### Clinical and immunological characteristics of the participants

According to their medical records obtained at the ART clinics, most of the participants, 549 (69.8%), were on first-line treatment and 237 (30.2%) were on second-line treatment. Participants had received services from an ART clinic for an average of 102 (SD = 74) months and had been on ART for 94 (SD = 70) months. The mean initial CD4 count was 580(SD = 948) cells/mm3. The mean viral load at first test was 27618 (SD = 133295) copies/ml, and at latest test was 13599 (SD = 86587) copies/ml. Based on the self-reported visual adherence scale, 92% were adherent to ART (Table 2). The rest of the clinical and immunological characteristics are presented in Table 2.

### Awareness and disclosure of HIV status

As shown in Table 3, most of the participants in this study, 702 (89.3%), knew that they were living with HIV, and 402(51.1%) reported that they acquired it from their mothers. About three quarters, 596 (75.8%) knew the kind of medicines they were receiving. Less than half of the participants, 379 (48.2%), had disclosed their HIV status to someone.

### The preparation process for the transition into adult care

According to this study, 247(31.4%) of the participants had received counselling on the transition to adult services, and of these, the majority 178(72.1%) were counselled by /peer educators and health providers 61(24.7%). However, only 33(4.2%) had completed a transfer form. In addition, only 97(12.3%) had visited an adult ART clinic to prepare for the transition, and of these majority, 40(41.2%) and 40(41.2%) were taken to the adult clinic by counsellors and peer educators, respectively. Almost all of the adolescents who had visited the adult Clinic 93 (95.9%) stated that the visit helped prepare for the transition. When asked about their preparedness to manage their treatment going forward, 445(56.6%) said that they were very prepared; 184(23.4%) were somewhat prepared, and 157(20.1%) were very unprepared. However, when asked about their satisfaction with the preparation process for transition in general, 126 (16.0%) reported that they were very satisfied; 226(28.8%) were somewhat satisfied; 399 (50.8%) were somewhat dissatisfied, and 35(4.5%) were very dissatisfied. The rest of the results on the experience for preparation for transition among participants in this study are presented in Table 4.

### Assessment of adolescents' readiness for the transition into adult care

Based on the readiness to transition tool employed in this study, 51 (6.5%) of the adolescents were ready to transition to adult care.

### Factors associated with readiness to transition of adolescents into adult care

Bivariate analyses showed that adolescents who were ready to transition were significantly more likely to be older (17.3±4 years vs. 19.4 ± 2 years, $p$ = 0.002), working for pay (21.9% vs. 43.1%, $p$ = 0.001), and their family having received social support for their health (29.3% vs.

**Table 1. Socio-demographic characteristics of the participants and their transition readiness.**

| Socio-demographic characteristic | Total (n = 786) | Readiness to transition | | |
|---|---|---|---|---|
| | n (%) | Not ready to transition n = 735 n (%) | Ready to transition n = 51 n (%) | P-value |
| **Age (in years)** | 17.48 ± 4 | 17.35 ± 4.1 | 19.35 ± 2.4 | **0.002** |
| **Gender** | | | | |
| Male | 302(38.4) | 284(38.6) | 18(35.3) | 0.635 |
| Female | 484(61.6) | 451(61.4) | 33(64.7) | |
| **Currently living with** | | | | |
| Parents | 362(46.1) | 336(45.7) | 26(51.0) | 0.874 |
| Grand parents | 93(11.8) | 86(11.7) | 7(13.7) | |
| Relatives | 115(14.6) | 109(14.8) | 6(11.8) | |
| In orphanage | 3(0.4) | 3(0.4) | 0(0.0) | |
| Others | 213(27.1) | 201(27.3) | 12(23.5) | |
| **Mother still alive** | | | | |
| Yes | 509(64.8) | 480(65.3) | 29(56.9) | 0.222 |
| No | 277(35.2) | 255(34.7) | 22(43.1) | |
| **Mother's level of education** | | | | |
| No education | 46(9.1) | 42(8.8) | 4(13.8) | 0.532 |
| Primary education | 150(29.5) | 144(30.1) | 6(20.7) | |
| Secondary education | 113(22.2) | 104(21.7) | 9(31.0) | |
| Tertiary education | 44(8.7) | 41(8.6) | 3(10.3) | |
| Don't know | 155(30.5) | 148(30.9) | 7(24.1) | |
| **Father still alive** | | | | |
| Yes | 417(53.1) | 393(53.5) | 24(47.1) | 0.375 |
| No | 369(46.9) | 342(46.5) | 27(52.9) | |
| **Main daily caregiver** | | | | |
| Parents | 360(45.8) | 334(45.4) | 26(51.0) | 0.784 |
| Grandparents | 100(12.7) | 92(12.5) | 8(15.7) | |
| Relatives | 107(13.6) | 101(13.7) | 6(11.8) | |
| In orphanage | 3(0.4) | 3(0.4) | 0(0.0) | |
| Others | 216(27.5) | 205(27.9) | 11(21.6) | |
| **Your level of education** | | | | |
| No education | 363(46.2) | 352(47.9) | 11(21.6) | **<0.001** |
| Primary education | 310(39.4) | 284(38.6) | 26(51.0) | |
| Secondary education | 43(5.5) | 37(5.0) | 6(11.8) | |
| Tertiary education | 48(6.1) | 40(5.4) | 8(15.7) | |
| Don't know | 22(2.8) | 22(3.0) | 0(0.0) | |
| **Are you employed** | | | | |
| Yes | 183(23.3) | 161(21.9) | 22(43.1) | **0.001** |
| No | 603(76.6) | 573(78.1) | 29(56.9) | |
| **The family received social support** | | | | |
| Yes | 240(30.5) | 215(29.3) | 25(49.0) | **0.003** |
| No | 546(69.5) | 520(70.7) | 26(51.0) | |

49.0%, *p* = 0.003). In contrast, adolescents who were not ready to transition were more likely to have no education (47.9% vs. 21.6%, *p* = 0.001) (**Table 1**).

Adolescents who were not ready to transition were significantly more likely to report that they did not know their disease (11.4% vs. 0.0%, *p* = 0.011), to not know how they got infected (32.0% vs. 11.8%, *p* = 0.007) and do not know the kind of medicines they were receiving

**Table 2. Clinical and immunological characteristics of adolescents living with HIV and transition readiness.**

| Variable | Total (n = 786) | Readiness to transition | | |
|---|---|---|---|---|
| | n (%) | Not ready to transition n = 735 n (%) | Ready to transition n = 51 n (%) | P-value |
| **Type of ART site** | | | | |
| Paediatric | 8 (1.0) | 8 (1.1) | 0 (0.0) | **<0.001** |
| Adult | 90 (11.5) | 71 (9.7) | 19 (37.3) | |
| Other | 688 (87.5) | 656 (89.3) | 32 (62.7) | |
| **Type of current treatment** | | | | |
| First line | 549 (69.8) | 513 (69.8) | 36 (70.6) | 0.905 |
| Second line | 237 (30.2) | 222 (30.2) | 15 (29.4) | |
| Time since first ART clinic visit (in months) | $102 \pm 74$ | $102 \pm 74$ | $101 \pm 73$ | 0.943 |
| Time since first ART initiation (in months) | $94 \pm 70$ | $94 \pm 69$ | $91 \pm 70$ | 0.767 |
| Time from first visit to ART start (in months) | $42 \pm 72$ | $40 \pm 71$ | $61 \pm 77$ | **0.044** |
| First CD4 | $580 \pm 948$ | $596 \pm 981$ | $384 \pm 320$ | 0.073 |
| First viral load | $27618 \pm 133295$ | $27807 \pm 134947$ | $25087 \pm 110105$ | 0.896 |
| Latest viral load | $13599 \pm 86587$ | $14246 \pm 89633$ | $4989 \pm 17073$ | 0.451 |
| Visual adherence scale | $92 \pm 29$ | $92 \pm 30$ | $93 \pm 9$ | 0.694 |

(25.7% vs. 2.0%, p = < 0.001). On the other hand, adolescents who were ready to transition were more likely to have disclosed their HIV status to someone (46.7% vs. 70.6%, p = 0.001) and to know that they were receiving ART (74.3% vs. 98.0%, $p = < 0.001$) (**Table 3**).

Adolescents who were ready to transition were significantly more likely to prefer receiving HIV treatment and care from Adult ART services (13.9% vs. 39.2%, p = <0.001), to have received counselling on transition to adult services (29.0% vs. 66.7%, p = < 0.001), to have completed a transfer form (3.4% vs. 15.7%, p = < 0.001) and to have visited an adult clinic to

**Table 3. Awareness and HIV status disclosure among adolescents living with HIV and their readiness to transition into adult ART services.**

| HIV knowledge and disclosure | Total (n = 786) n (%) | Readiness to transition | | |
|---|---|---|---|---|
| | | Not ready to transition n = 735 n (%) | Ready to transition n = 51 n (%) | P-value |
| **Can you tell us what your disease is?** | | | | |
| HIV/AIDS | 702 (89.3) | 651 (88.6) | 51 (100.0) | **0.011** |
| Don't know | 84 (10.7) | 84 (11.4) | 0 (0.0) | |
| **Do you know how you got infected?** | | | | |
| Mother to child | 402 (51.1) | 371 (50.5) | 31 (60.8) | **0.007** |
| Don't know | 241 (30.7) | 235 (32.0) | 6 (11.8) | |
| Others | 143 (18.2) | 129 (17.6) | 14 (27.5) | |
| **Do you know how this disease is transmitted?** | | | | |
| Mother to child | 77 (9.8) | 72 (9.8) | 5 (9.8) | **0.038** |
| Unprotected sex | 397 (50.5) | 361 (49.2) | 36 (70.6) | |
| Sharing needles | 57 (7.3) | 56 (7.6) | 1 (2.0) | |
| Blood transfusion | 5 (0.6) | 5 (0.7) | 0 (0.0) | |
| Others | 248 (31.6) | 239 (32.6) | 9 (17.6) | |
| **Do you know what kind of medicine you have received?** | | | | |
| ART | 596 (75.8) | 546 (74.3) | 50 (98.0) | **<0.001** |
| Don't know | 190 (24.2) | 189 (25.7) | 1 (2.0) | |
| **Have you ever disclosed your HIV status to anyone?** | | | | |
| Yes | 379 (48.2) | 343 (46.7) | 36 (70.6) | **0.001** |
| No | 407 (51.8) | 392 (53.3) | 15 (29.4) | |

**Table 4. Experience of preparation for the transition of adolescents to adult ART care and transition readiness.**

| Experience of preparation for transition | (n = 786) n (%) | Not ready to transition n = 735 n (%) | Ready to transition n = 51 n (%) | P-value |
|---|---|---|---|---|
| **Total Readiness to transition** | | | | |
| **Which facility do you prefer to receive HIV treatment and care?** | | | | |
| Pediatric ART services | 48(6.1) | 48(6.5) | 0(0.0) | <**0.001** |
| Adult ART services | 122(15.5) | 102(13.9) | 20(39.2) | |
| Adolescent friendly services | 616(78.4) | 585(79.6) | 31(60.8) | |
| **The person you trust most with your treatment?** | | | | |
| Health care provider | 261(33.2) | 248(33.8) | 13(25.5) | **0.018** |
| Counsellor | 55(7.0) | 49(6.7) | 6(11.8) | |
| Peer educator | 11(1.4) | 9(1.2) | 2(3.9) | |
| Friends | 33(4.2) | 27(3.7) | 6(11.8) | |
| Family | 389(49.5) | 368(50.2_ | 21(41.2) | |
| Others | 35(4.5) | 32(4.4) | 3(5.9) | |
| **Received counselling on the transition to adult services** | | | | |
| Yes | 247(31.4) | 213(29.0) | 34(66.7) | <**0.001** |
| No | 539(68.6) | 522(71.0) | 17(33.3) | |
| **The person who provided the counselling** | | | | |
| Health provider | 61(24.7) | 53(24.9) | 8(23.5) | 0.495 |
| Counsellor/ peer educator | 178(72.1) | 152(71.4) | 26(76.5) | |
| Others | 8(3.2) | 8(3.8) | 0(0.0) | |
| **Ever completed a transfer form** | | | | |
| Yes | 33(4.2) | 25(3.4) | 8(15.7) | <**0.001** |
| No | 753(95.8) | 710(96.6) | 43(8.3) | |
| **Ever visited an adult clinic to prepare for the transition** | | | | |
| Yes | 97(12.3) | 66(9.0) | 31(60.8) | <**0.001** |
| No | 689(87.7) | 669(91.0) | 20(39.2) | |
| **The person who took you to the adult clinic to prepare for the transition** | | | | |
| Counsellors | 40(41.2) | 28(42.4) | 12(38.7) | 0.335 |
| Peer educators | 40(41.2) | 24(36.4) | 16(51.6) | |
| Friends | 1(1.0) | 1(1.5) | 0(0.0) | |
| Family | 6(6.2%) | 6(9.1) | 0(0.0) | |
| Others | 10(10.3) | 7(10.6) | 3(9.7) | |
| **Was the visit helpful to cope with the transition** | | | | |
| Yes | 93(95.9) | 62(93.9) | 31(100.0) | 0.162 |
| No | 4(4.1) | 4(6.1) | 0(0.0) | |
| **Is there a person in the identified to support you during the transition** | | | | |
| Yes | 76(78.4) | 48(72.7) | 28(90.3) | 0.050 |
| No | 21(21.6) | 18(27.3) | 3(9.7) | |
| **Were you prepared to manage your treatment going forward?** | | | | |
| Very prepared | 445(56.6) | 398(54.1) | 47(92.2) | <**0.001** |
| Somewhat prepared | 184(23.4) | 182(24.8) | 2(3.9) | |
| Very unprepared | 157(20.0) | 155(21.1) | 2(3.9) | |
| **Satisfied with the preparation process for transition in general** | | | | |
| Very satisfied | 126 (16.0) | 95 (12.9) | 31 (60.8) | <**0.001** |
| Somewhat satisfied | 226 (28.8) | 218 (29.7) | 8 (15.7) | |
| Somewhat dissatisfied | 399 (50.8) | 388 (52.8) | 11 (21.6) | |
| Very dissatisfied | 35 (4.5) | 34 (4.6) | 1 (2.0) | |

prepare for transition (9.0% vs. 60.8%, p = < 0.001). In addition, adolescents who were ready to transition perceived that they were very prepared to manage their treatment going forward (54.1% vs. 92.2%, *p* = < 0.001), and reported that they were very satisfied with the preparation process for transition in general (12.9% vs. 60.8%, *p* < 0.001) (**Table 4**).

Multivariate logistic regression analysis found that readiness to transition into adult care remained significantly associated with having acquired a tertiary education (AOR 4.535, 95% CI 1.243–16.546, *P* = 0.022), trusting peer educators for HIV treatment (AOR 16.222, 95% CI 1.835–143.412, P = 0.012), having received counselling on transition to adult services (AOR 2.349, 95% CI 1.004–5.495, *P* = 0.049), having visited an adult clinic to prepare for transition (AOR 6.616, 95% CI 2.435–17.987, P = < 0.001) and being satisfied with the transition process in general (AOR 0.213, 95% CI 0.069–0.658, P = 0.007) (**Table 5**).

## Discussion

Ensuring effective transition from pediatric/adolescent/young people to adult care is a countrywide priority for optimizing young people's health and critical for the prevention of HIV transmission to wider communities. Also, understanding the factors associated with adolescents' transition readiness from adolescent clinics to adult clinics is critical in designing adolescent-friendly services that will facilitate successful transitioning from adolescent to adult care. However, there is limited information about transitioning adolescents to adult care and its associated factors in Uganda. Hence, this study aimed to assess the factors associated with adolescents' transition readiness from adolescent clinics to adult ART clinics.

**Table 5. Factors associated with readiness to transition of adolescents from adolescent clinics to adult ART clinic in the multivariable logistic regression model.**

| Variables in the final model | Readiness to transition | |
|---|---|---|
| | AOR (95% CL) | P-value |
| **Your level of education** | | |
| No education | Reference | |
| Primary education | 1.345 (0.547–3.309) | 0.518 |
| Secondary education | 1.322 (0.352–4.965) | 0.679 |
| Tertiary education | 4.535 (1.243–16.546) | **0.022** |
| **The person you trust most with your treatment** | | |
| Health provider | Reference | |
| Counsellor | 2.208 (0.594–8.199) | 0.237 |
| Peer educator | 16.222 (1.835–143.412) | **0.012** |
| Friends | 0.873 (0.192–3.974) | 0.861 |
| Family | 0.820 (0.329–2.044) | 0.670 |
| Others | 0.865 (0.175–4.283) | 0.859 |
| **Received counselling on the transition to adult services** | | |
| No | Reference | |
| Yes | 2.349 (1.004–5.495) | **0.049** |
| **I visited an adult clinic to prepare for the transition** | | |
| No | Reference | |
| Yes | 6.619 (2.435–17.987) | **<0.001** |
| **Satisfied with the preparation process for transition in general** | | |
| Very satisfied | Reference | |
| Somewhat satisfied | 0.216 (0.012–3.805) | 0.295 |
| Somewhat dissatisfied | 0.546 (0.179–1.670) | 0.289 |
| Very dissatisfied | 0.213 (0.069–0.658) | **0.007** |

Adolescents are a very heterogeneous group, and that age is a critical variable related to the sexual, physical, sexual, cognitive, and psychological development of the adolescents. We also acknowledge that adolescents of the same age may be at different levels of development. Indeed, some of the adolescents live on their own; some are already parents, while others are dependent on parents or guardians. While the preparation of transitioning should start early as per standard operating procedures, to ensure that by the time adolescents reach the age of 18 years, where they have to nominally transfer to the adult's HIV clinic, it is not clear that by that age, all adolescents are ready and prepared for the HIV care transition. The fact that there are some young people still seeking care in the adolescent HIV clinics, as well as older adolescents who seek care from adult HIV clinics, is an indication that many individuals may not be ready for or may experience some challenges and barriers in the HIV care transition. Factors associated with readiness to transition included; peer education, education level, receiving counselling on the transition to adult care, and visiting an adult clinic to prepare for the transition.

Early preparation of adolescents and young people living with HIV for transition is essential for the effective transition to adult care, and according to the guidelines, it should be a process that starts early, even before adolescence. The overall readiness for transition from paediatric/adolescent to adult ART care among YPLHIV in this study was very low at 51 (6.5%). This could be attributed to the fact that the majority of the participants in this study were below 20 years. Yet, most of the facilities in this study reported starting to prepare their adolescents and young people for transitioning at 20 years. They are ready by 24 years and therefore were not yet prepared for transitioning. This study's level of transition readiness was lower than that earlier found in a study conducted in Cambodia, which found that 53.3% of the adolescents had a high level of transition readiness to adult ART care [15]. This difference could be due to the implementation of the transitioning process in different countries.

Adolescents with HIV have been found to develop strong and long-lasting relations with their care teams and fellow HIV adolescents, hence seeing them as their extended family members [16, 17]. A high level of transition readiness to adult care was independently associated with trust in peer educators for HIV treatment in our study. These findings are consistent with those of other studies that have also shown the importance of peers and friends coping with adult care transition [15, 18]. Peer support can help adolescents develop confidence in themselves and also acquire different coping mechanisms and skills.

A high level of readiness to transition was independently associated with having tertiary education. These findings are in line with those of other studies, which also found that attaining a higher education was a strong predictor of uptake of HIV related services, including voluntary counselling and testing [19, 20], prevention of mother to child transmission of HIV [21], and transition to adult care [22]. This may be attributed to the fact that HIV-positive adolescents who have attained higher education have more knowledge about their disease and self-care management skills. Our findings suggest that some of the challenges for adolescents for a transition readiness include leaving people they are familiar and comfortable with when they transit to adult HIV clinics, being seen by healthcare providers who are unfamiliar with their illness, challenges of change due to growing up, leaving a familiar clinic and staff whom they trust, and losing the ongoing support system which they have enjoyed over the years in the adolescent clinic, and the reluctance of healthcare providers in the adolescent clinic to facilitate a smooth transition process.

Counselling on the transition to adult care is critical in preparing adolescents and young people for a better transition process. Our study's high level of transition readiness was independently associated with receiving specific counselling on the transition to adult services while still in adolescent clinics. Emphasizing the need and counselling the adolescents about transitioning is a facilitator for transitioning, also reported in other studies [23, 24]. Several

adolescents have stated inbuilt fears about transitioning, including fear to lose friends and peers when transitioned to adult art care, fear of stigma and discrimination while in the adult clinics [24, 25], and therefore addressing such fears through counselling while in the adolescent clinics will enable these adolescents to address and overcome such fears and prepare them for a better transition.

Creating familiarity with the adult clinics, such as scheduling visits to these clinics, has a significant effect in facilitating adolescents' connection to the adult clinics and the overall transition in general [15, 26]. Scheduling visits to the adult clinic before transition enables adolescents to become oriented to the new clinic setting, health providers, and expectations [27–29]. These findings are consistent with our study results, which showed that a higher level of transition readiness was independently associated with having visited an adult clinic to prepare for the transition. Therefore, visiting the adult clinic enables these adolescents to acquire insight into what happens in the adult clinic to get to know the different health providers in the adult clinic and get familiar with the adults themselves.

While of much importance, age is not the only important predictor of transition readiness, and age may not correspond with physical, cognitive, sexual, or psychological maturity, all of which may influence transition readiness. While most HIV- infected adolescents transition to adult care is usually between 22–24 years of age, there are some who show readiness at earlier or later ages. From our findings, a stratified analysis of transitional readiness showed that compared to adolescents 10–14 years, older adolescents and young people were more likely to manifest transitional readiness. However, chronological age is not the only predictor of transitional readiness. Growing adolescents face unique developmental, psychological, and sexuality challenges as they are still maturing physically, mentally, and sexually. Such adolescents with chronic illnesses often engage in many of the same risk behaviors as their unaffected peers, which behaviors put them at risk of increased morbidity and mortality, and with the potential to affect readiness, preparedness, and coping ability for the HIV care transition. This is in addition to contextual challenges encountered by most people living with HIV regardless of age, including stigma, disclosure difficulties, challenges with adherence, and socioeconomic hardships, relationship issues, and sexuality problems and all of which prevent attainment of optimal health outcomes in case they affect transition readiness or eventually successful transition. Our findings further confirm that transitional readiness is multifaceted and impacts the access to medical care. Also, the findings suggest that the timing of the transition may differ for each individual in a given clinical and social context. While it may be dependent on factors such as age, developmental stage of the adolescent or youth, and other social factors, it is possibly very challenging for some adolescents.

## Conclusion

Preparation for transition to adult care for adolescents living with HIV needs to be a planned process involving counselling and adolescents' exposure to what goes on in adult HIV clinics. Counselling on the transition to adult care is critical in preparing adolescents and young people for a better transition process and ensure transition readiness. This counselling may need to be started early. It should be individualized so that it is personalised to adolescents' specific needs, as transition readiness in our study was independently associated with receiving specific counselling on the transition to adult services while still in adolescent clinics. Such counselling should address barriers to transition, such as fear of losing friends and peers when transitioned to adult art care, fear of stigma and discrimination in adult clinics, privacy, and confidentiality. Future research may be needed to assess the quality of care for adolescents after transition and assess the effectiveness of different models of adolescent transition.

## Supporting information

**S1 Dataset. Anonymized data set.**
(XLSX)

**S1 File. Ministry of health transition tool.**
(DOCX)

## Acknowledgments

The authors would like to thank the adolescents and young people in all the facilities we collected data and the research assistants.

## Author Contributions

**Conceptualization:** Scovia Nalugo Mbalinda, Sabrina Bakeera-Kitaka, Mathew Nyashanu, Dan Kabonge Kaye.

**Data curation:** Scovia Nalugo Mbalinda, Derrick Amooti Lusota.

**Formal analysis:** Scovia Nalugo Mbalinda, Derrick Amooti Lusota, Dan Kabonge Kaye.

**Funding acquisition:** Scovia Nalugo Mbalinda, Philippa Musoke.

**Investigation:** Scovia Nalugo Mbalinda.

**Methodology:** Scovia Nalugo Mbalinda, Sabrina Bakeera-Kitaka, Derrick Amooti Lusota, Philippa Musoke, Dan Kabonge Kaye.

**Project administration:** Scovia Nalugo Mbalinda.

**Resources:** Scovia Nalugo Mbalinda.

**Software:** Scovia Nalugo Mbalinda, Derrick Amooti Lusota.

**Supervision:** Scovia Nalugo Mbalinda, Sabrina Bakeera-Kitaka, Derrick Amooti Lusota, Dan Kabonge Kaye.

**Validation:** Scovia Nalugo Mbalinda.

**Visualization:** Scovia Nalugo Mbalinda.

**Writing – original draft:** Scovia Nalugo Mbalinda, Sabrina Bakeera-Kitaka, Derrick Amooti Lusota, Mathew Nyashanu.

**Writing – review & editing:** Scovia Nalugo Mbalinda, Sabrina Bakeera-Kitaka, Derrick Amooti Lusota, Philippa Musoke, Mathew Nyashanu, Dan Kabonge Kaye.

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
