## [Decision Letter · Decision Letter 0]

9 Feb 2021

PONE-D-21-00972

Transition to adult care:  Exploring factors associated with transition readiness among adolescents and young people in adolescent ART clinics in Uganda

PLOS ONE

Dear Dr. Scovia,

Thank you for submitting your manuscript to PLOS ONE. After careful consideration, we feel that it has merit but does not fully meet PLOS ONE’s publication criteria as it currently stands. Therefore, we invite you to submit a revised version of the manuscript that addresses the points raised during the review process.

We look forward to receiving your revised manuscript.

Kind regards,

Claudia Marotta

Academic Editor

PLOS ONE

Additional Editor Comments (if provided):

dear authors follow reviewer suggestions to improve your article

Journal Requirements:

2) In the Methods section, please provide a justification for the stratification of readiness to transition score used in the study (ie, please provide a justification as to why scores of 30< were classified as ready to be transitioned).

3) Please include additional information regarding the survey or questionnaire used in the study and ensure that you have provided sufficient details that others could replicate the analyses. For instance, if you developed a questionnaire as part of this study and it is not under a copyright more restrictive than CC-BY, please include a copy, in both the original language and English, as Supporting Information.

4) We suggest you thoroughly copyedit your manuscript for language usage, spelling, and grammar. If you do not know anyone who can help you do this, you may wish to consider employing a professional scientific editing service.  

5)  We note that the grant information you provided in the ‘Funding Information’ and ‘Financial Disclosure’ sections do not match.

6) Please amend your list of authors on the manuscript to ensure that each author is linked to an affiliation. Authors’ affiliations should reflect the institution where the work was done (if authors moved subsequently, you can also list the new affiliation stating “current affiliation:….” as necessary).

7) Please include captions for your Supporting Information files at the end of your manuscript, and update any in-text citations to match accordingly. Please see our Supporting Information guidelines for more information: http://journals.plos.org/plosone/s/supporting-information.

Reviewers' comments:

Reviewer's Responses to Questions

**Comments to the Author**

1. Is the manuscript technically sound, and do the data support the conclusions?

Reviewer #1: No

Reviewer #2: Yes

2. Has the statistical analysis been performed appropriately and rigorously? 

Reviewer #1: Yes

Reviewer #2: Yes

3. Have the authors made all data underlying the findings in their manuscript fully available?

Reviewer #1: Yes

Reviewer #2: Yes

4. Is the manuscript presented in an intelligible fashion and written in standard English?

Reviewer #1: Yes

Reviewer #2: Yes

5. Review Comments to the Author

Reviewer #1: Transition to adult care: Exploring factors associated with transition readiness among adolescents and young people in adolescent ART clinics in Uganda

Study Summary

The study of Mbalinda and colleagues aims at exploring factors associated with adolescent readiness for the transition into adult care in Uganda. To achieve this goal, the authors conducted a cross-sectional study involving 786 individuals, aged 10 years to 24 years, among whom they explored the readiness for transition to adult clinics using structured interviews. The results indicated that 51 (6.5%) of the participants “adolescents” were ready to transition to adult care, based on the readiness to transition tolls they employed. The authors discussed their findings around this low readiness to transition in the ART program in Uganda.

Reviewer observations

Children and adolescent currently represent key populations with increasing challenges regarding HIV/AIDS care and monitoring management. It is obvious that specific strategies should be implemented for those key populations and should be adapted to their specific needs. Transition from pediatric services to adult wards is an important step, which may significantly affect the future outcome of ART of the adolescent if not adequately managed. This highlights the importance of such investigations that may help identifying factors that can affect the success of the transition.

Major queries

A. The study population

According to WHO, adolescents refer to individuals aged 10 to 19 years. This population is managed in pediatric/adolescent services until they reach the age for transition to adult services.

The study population of this report includes both adolescents and young adults aged from 10 to 24 years. It not clear in this study design and it is not indicated anywhere in the manuscript if the age group considered in Uganda for adolescent is 10-24 years. I believe not, since the authors consider the group from 20 to 24 years old as “young adults” and not adolescents.

In addition, in the methods presented, there is no indication if this study population “adolescent and young adults” was recruited in pediatric/adolescent services or in adult services as well. It is only mentioned (lines 120-121) that they were recruited “ …in antiretroviral therapy (ART) clinics in four regions of Uganda”.

This is a critical aspect of the validity of this study, since adequate definition of the population studied is essential to achieve the expected outcome. Is not clear how assessing transition readiness among a so wide age group (10 to 24) using the same investigation tools/questionnaires, will permit identifying factors affecting the transition. Are individuals aged 10, 11, 12, 13, 14 years, etc, expected to be ready for transition to adult services?

The high heterogeneity of this population can be the main reason of the very low transition readiness that the authors reported 51/786 (6.5%).

There is no rational on the selection of the different aged groups:

- What was expected as outcome from the population of 10-14? Is it expected that adolescent of this age group are ready for the transition?

- For those aged 20-24 years, are they still follow-up in pediatric/adolescent services? If no, why was this population included in the study?

- If the age of transition to adult services in Uganda is 24 years, in this case even for the group of 15-19 years, it will be hard to expect the same perception of readiness to transition compared to those aged 20-24 years.

This key aspect of the study, in my opinion, critically affects the validity of the results obtained.

B. Other key points

- Lines 80-88: the characteristics presented as referring to “transition readiness” are not supported by any reference. Are these definitions made by the authors, Ugandan authorities or international organizations? No reference presented.

- Lines 100-101: “…Further, those at the highest risk for health complications and transmission of HIV to others (males, those living further away from health facilities) are…”. Do authors have specific references to support this?

Reviewer #2: Line 66/67:delete redundant In Uganda [2]. Line 120 did you mean PLHIV? Also please spell out PLHIV when used for the first time in the manuscript. For lines 255-259, recommend listed the statistic for those ready for transition first vs those not ready for transition as opposed to the other way round. eg. bivariate analyses showed that adolescents who were ready to transition were significantly more like to be older (19.4+-2 yrs vs 17.3+-4 yrs).

6. PLOS authors have the option to publish the peer review history of their article (what does this mean?). If published, this will include your full peer review and any attached files.

Reviewer #1: No

Reviewer #2: No

---

## [Author Response · Author response to Decision Letter 0]

22 Feb 2021

Makerere University

 P.O. Box 7072 Kampala, Uganda Tel: 256 414 530404

 E-mail: nursing@chs.mak.ac.ug

SCHOOL OF HEALTH SCIENCES

Department of Nursing

22nd February 2021 

The Editor, 

PLOS ONE 

Dear Sir.

Re: Response to Reviewers comments 

 Thank you for your encouragement to revise and to resubmit our manuscript entitled: "Transition to adult care: Exploring factors associated with transition readiness among adolescents and young people in adolescent ART clinics in Uganda." 

Attached is our resubmission.

 We have attempted to address all the reviewers' comments and recommendations. We believe that our manuscript has been strengthened, and we thank the reviewers for taking the time to read the manuscript and to specify these recommendations

Yours faithfully,

ScoviaNalugoMbalinda

Makerere University, College of Health Sciences, Department of Nursing

P.O. Box 7072, Kampala, Uganda

E-mail: smbalinda@gmail.com.

Comments 

Major queries

A. The study population

1. According to WHO, adolescents refer to individuals aged 10 to 19 years. This population is managed in pediatric/adolescent services until they reach the age for the transition to adult services.

Response: Yes, adolescents are aged 10-19, but when we went to ART clinics, there were young people (aged above 19 years) who were still attending the clinic for HIV care services. Secondly, some adolescents are aged 19 years, and less attend HIV care in the adult HIV clinic. 

2. The study population of this report includes both adolescents and young adults aged from 10 to 24 years. It not clear in this study design and it is not indicated anywhere in the manuscript if the age group considered in Uganda for adolescent is 10-24 years. I believe not, since the authors consider the group from 20 to 24 years old as "young adults" and not adolescents.

Response: Adolescents are aged 10-19, but we found that the adolescents had refused to go to adult ART clinics. We included adolescents (10-19 years) and young people to capture the age group 20-24. That why we included young people in the topic and the methodology we have stated that adolescents and young people. Page line 126,130 on page 6. 

The provision of HIV care in this period is a continuum. The anomaly is that some adolescents attend HIV care in adult clinics while some young people still seek HIV care in adolescent clinics. Both scenarios potentially affect the quality of care received. That's why in the topic, we included adolescents and young people in the study to analyse the transition of HIV care from adolescent to adult clinics, with a focus on transition readiness from adolescent to adult HIV care. The questions where information was needed are whether, irrespective of age, the population of adolescents and young people were ready to transition from adolescent to adult clinics, as were their readiness for the transition process. Line 113-116, page 6.

3. In addition, in the methods presented, there is no indication if this study population "adolescent and young adults" was recruited in pediatric/adolescent services or in adult services as well. It is only mentioned (lines 120-121) that they were recruited "…in antiretroviral therapy (ART) clinics in four regions of Uganda".

Response: The adolescents and young people were recruited from adolescent ART clinics in each facility. The adolescent clinics were running once a week. Line 130, Page 6. 

The participants' selection was the age of the adolescents and young people who were attending HIV care in the adolescent clinics. 

4. This is a critical aspect of the validity of this study since an adequate definition of the population studied is essential to achieve the expected outcome. It is not clear how assessing transition readiness among a so wide age group (10 to 24) using the same investigation tools/questionnaires will identify factors affecting the transition. Are individuals aged 10, 11, 12, 13, 14 years, etc, expected to be ready for transition to adult services? 

Response: The preparation of transitioning starts early as per standard operating procedures.

It starts 10-13 years by making transition plans and addressing concerns about support for parents/ care during transitions. As adolescents grow, they add other issues of independence and responsibility for themselves. Every 3-6 months, the adolescents are assessed as per the protocol described in the paper. If they score 30 and above, then they are ready to transition.

The age of transition: Most HIV- infected adolescents transition to adult care between 22 and 24 years of age. However, developmental stages and readiness for transition may be better indicators than chronological age for determining when the transition should occur. Patients with developmental delays or chaotic and unstable life may need more time to become ready to transition. Adolescents who demonstrate independence in making their own decision and show responsibility for their care may be ready sooner.

We had earlier explained the process in line 104-112, page 5-6

5. The high heterogeneity of this population can be the main reason of the very low transition readiness that the authors reported 51/786 (6.5%).

Response: Yes, that could be the reason, however like we said, the transition is a process, and it starts as early as ten years, and the adolescents are assessed on different aspects like HIV knowledge and disclosure, Transition process, Knowledge of own health, Knowledge of responsible behavior, Knowledge of response of emergency care, Knowledge of how to manage health care needs, Demonstration of responsible sexual behavior, keeping track of health needs, support groups, and transition plan. Each adolescent is assessed in an individualized manner for skills development and understanding of successful transition and Address barriers for each patient that may prevent skills acquisition and providers from developing a transition plan and then implementing that plan.

6. There is no rational on the selection of the different aged groups:

- What was expected as outcome from the population of 10-14? Is it expected that adolescent of this age group are ready for the transition?

Response: Transition process is a process. It does not start at the age of transition but as early as ten years. So irrespective of the age as long you are ready for the transition as per protocol you will be transitioned.

- For those aged 20-24 years, are they still follow-up in pediatric/adolescent services? If no, why was this population included in the study?

Response: Yes, this group was in the adolescent clinic, and they had failed to transition. That is why they were included in the study 

- If the age of transition to adult services in Uganda is 24 years, in this case even for the group of 15-19 years, it will be hard to expect the same perception of readiness to transition compared to those aged 20-24 years.

Response: Initially, the transition age was 19. However, the Ministry of health realized that some of the adolescents were not ready for the transition. It increased the age of transition to adult care between 22-24 years of age. But protocol stated that developmental stages and readiness for transition might be better indicators than chronological age for determining when the transition should occur. Patients with developmental delays or chaotic and unstable life may need more time to become ready to transition. Adolescents who demonstrate independence in making their own decision and show responsibility for their own care would transition earlier.

B. Other key points

- Lines 80-88: the characteristics presented as referring to "transition readiness" are not supported by any reference. Are these definitions made by the authors, Ugandan authorities, or international organizations? No reference presented.

Response: Reference is added. Line 88, page 5

- Lines 100-101: "…Further, those at the highest risk for health complications and transmission of HIV to others (males, those living further away from health facilities) are…". Do authors have specific references to support this?

Response: The references are there 11-13

Journal Requirements:

Response: This has been addressed in the manuscript

2) In the Methods section, please provide a justification for the stratification of readiness to transition score used in the study (ie, please provide a justification as to why scores of 30< were classified as ready to be transitioned).

Response: The score was used because it's a score that the Ministry of Health uses as a cut-off to show whether the adolescent or young person is ready for the transition. The total score of the tool was 38. If the young person scored 30 and above, they were ready to be transitioned. If they scored 25- 30, they would be reassessed and transitioned after three months in case they score 30. I have added the reference for this line 158, page 8. I have also attached the tool as a supporting file. 

3) Please include additional information regarding the survey or questionnaire used in the study and ensure that you have provided sufficient details that others could replicate the analyses. For instance, if you developed a questionnaire as part of this study and it is not under a copyright more restrictive than CC-BY, please include a copy, in both the original language and English, as Supporting information.

Response: Ministry of Health developed the tool to aid the transition in the clinical area. We only added the socio-demographic and clinical, and immunological data. We have attached it as a supporting file 

4) We suggest you thoroughly copyedit your manuscript for language usage, spelling, and grammar. If you do not know anyone who can help you do this, you may wish to consider employing a professional scientific editing service. 

Response: We have edited the manuscript 

 Response: This has been addressed 

5) We note that the grant information you provided in the 'Funding Information' and 'Financial Disclosure' sections do not match.

When you resubmit, please ensure that you provide the correct grant numbers for the awards you received for your study in the 'Funding Information' section.

Response: This has been addressed

6) Please amend your list of authors on the manuscript to ensure that each author is linked to an affiliation. Authors' affiliations should reflect the institution where the work was done (if authors moved subsequently, you can also list the new affiliation stating "current affiliation:…." as necessary).

Response: This has been addressed

7) Please include captions for your Supporting Information files at the end of your manuscript, and update any in-text citations to match accordingly. Please see our Supporting Information guidelines for more information: http://journals.plos.org/plosone/s/supporting-information. 

Response: This has been addressed Page line 555, page 30

---

## [Decision Letter · Decision Letter 1]

29 Mar 2021

Transition to adult care:  Exploring factors associated with transition readiness among adolescents and young people in adolescent ART clinics in Uganda

PONE-D-21-00972R1

Dear Dr.  Mbalinda,

We’re pleased to inform you that your manuscript has been judged scientifically suitable for publication and will be formally accepted for publication once it meets all outstanding technical requirements.

Kind regards,

Claudia Marotta

Academic Editor

PLOS ONE

Additional Editor Comments (optional):

congratulations

Reviewers' comments:

Reviewer's Responses to Questions

**Comments to the Author**

1. If the authors have adequately addressed your comments raised in a previous round of review and you feel that this manuscript is now acceptable for publication, you may indicate that here to bypass the “Comments to the Author” section, enter your conflict of interest statement in the “Confidential to Editor” section, and submit your "Accept" recommendation.

Reviewer #2: All comments have been addressed

2. Is the manuscript technically sound, and do the data support the conclusions?

Reviewer #2: Yes

3. Has the statistical analysis been performed appropriately and rigorously? 

Reviewer #2: Yes

4. Have the authors made all data underlying the findings in their manuscript fully available?

Reviewer #2: Yes

5. Is the manuscript presented in an intelligible fashion and written in standard English?

Reviewer #2: Yes

6. Review Comments to the Author

Reviewer #2: Lines 327-329 is confusing, please rephrase.

Need to expand the thought in lines 331 and 332 some more: what is different in the implementation of the transitioning process in Uganda vs Cambodia

7. PLOS authors have the option to publish the peer review history of their article (what does this mean?). If published, this will include your full peer review and any attached files.

Reviewer #2: No

---

## [Editor Report · Acceptance letter]

12 Apr 2021

PONE-D-21-00972R1 

Transition to adult care:  Exploring factors associated with transition readiness among adolescents and young people in adolescent ART clinics in Uganda 

Dear Dr. Mbalinda:

I'm pleased to inform you that your manuscript has been deemed suitable for publication in PLOS ONE. Congratulations! Your manuscript is now with our production department. 

Kind regards, 

on behalf of

Dr. Claudia Marotta 

Academic Editor

PLOS ONE